# Biological Roles of Lipids in Rice

**DOI:** 10.3390/ijms25169046

**Published:** 2024-08-21

**Authors:** Kun Zhou, Zhengliang Luo, Weidong Huang, Zemin Liu, Xuexue Miao, Shuhua Tao, Jiemin Wang, Jian Zhang, Shiyi Wang, Xiaoshan Zeng

**Affiliations:** 1Hunan Rice Research Institute, Hunan Academy of Agricultural Sciences, Changsha 410125, China; zhzhkp@163.com (K.Z.); luozl@hunaas.cn (Z.L.); hwd4@163.com (W.H.); dhlzmin@163.com (Z.L.); miaoxuexue410@163.com (X.M.); taoshuhua2020@163.com (S.T.); wangjiemin2009@163.com (J.W.); 2State Key Lab of Rice Biology and Breeding, China National Rice Research Institute, Hangzhou 311400, China; 82101225170@caas.cn

**Keywords:** rice (*Oryza sativa* L.), lipids, genetic improvement, grain quality, pollen fertility, seed germination, seed longevity, biotic stress, abiotic stress, yield

## Abstract

Lipids are organic nonpolar molecules with essential biological and economic importance. While the genetic pathways and regulatory networks of lipid biosynthesis and metabolism have been extensively studied and thoroughly reviewed in oil crops such as soybeans, less attention has been paid to the biological roles of lipids in rice, a staple food for the global population and a model species for plant molecular biology research, leaving a considerable knowledge gap in the biological roles of lipids. In this review, we endeavor to furnish a current overview of the advancements in understanding the genetic foundations and physiological functions of lipids, including triacylglycerol, fatty acids, and very-long-chain fatty acids. We aim to summarize the key genes in lipid biosynthesis, metabolism, and transcriptional regulation underpinning rice’s developmental and growth processes, biotic stress responses, abiotic stress responses, fertility, seed longevity, and recent efforts in rice oil genetic improvement.

## 1. Introduction

Lipids are organic nonpolar molecules containing hydrogen, carbon, and oxygen atoms, which provide the framework for the structure and function of living cells. As a major type of lipid, triglycerides are stored as fat in adipose cells, serving as the high-density energy-storage depot for organisms and providing thermal insulation. Some lipids, like steroid hormones, act as chemical messengers between cells, tissues, and organs [1]. In terms of cellular function, lipids are integral components of cell membranes, contributing to their structure, flexibility, and permeability. They are also involved in cell signaling and the regulation of membrane-bound enzymes [2]. Plant-derived or vegetable oils are of significant importance to human society due to their versatile and wide-ranging applications. These oils are crucial dietary fats for human consumption and are extensively utilized in culinary practices and food production. They also serve as essential raw materials for the production of biofuels, offering sustainable alternatives to traditional fossil fuels. In the cosmetic and personal care industry, plant-derived oils are prized for their emollient and moisturizing properties, making them valuable ingredients in skincare products, soaps, and hair care formulations. Furthermore, these oils play a vital role in industrial processes, contributing to manufacturing paints, lubricants, and various commercial products [3]. Their diverse applications highlight the pivotal role of plant-derived oils in both consumer products and industrial processes.

Given the biological and economic importance of plant-derived lipids, the genetic pathways and regulatory networks of lipid biosynthesis and metabolism have been extensively studied and thoroughly reviewed, particularly in oil crops such as soybeans, rapeseed, and sunflowers [4,5,6]. Rice is a staple food for more than half of the world’s population, particularly in Asia. The lipids in rice grains comprise triglycerides (TAGs), phospholipids, fatty acids, and other bioactive lipid components, with a total amount ranging from 2 to 3%, much lower than oil crops [7]. The lipids are concentrated in rice embryos and brans, essentially determining milling, appearance, and cooking quality [7]. In addition, lipids are the essential building blocks of cellular membranes and act as signal molecules to modulate various biological processes [2]. Despite the full recognition of the significant roles of lipids in rice, less attention has been paid to lipids when compared to proteins and nucleic acids in the rice research community, leaving a considerable knowledge gap in the biological roles of lipids [8]. Fortunately, recent molecular genetic research has uncovered a number of rice genes involved in lipid biosynthesis, metabolism, and signaling pathways underpinning various biological processes. This review endeavors to furnish a current overview of the advancements in understanding the genetic foundation and physiological functions of lipids in the context of rice’s developmental and growth processes, biotic stress responses, abiotic stress responses, and fertility, as well as seed longevity (Figure 1; Table 1).

## 2. Roles of Lipids in Rice Growth and Development

In very-long-chain fatty acids (VLCFAs) biosynthesis, β-ketoacyl CoA synthase (E.C.2.3.1.119, KCS) is the key enzyme. Knock-out of WSL1 encoding a typical KCS protein resulted in pleiotropic phenotypes, including reduced growth, leaf fusion, sparse wax crystals, and low fertility. The observed phenotype might be attributed to the significant reduction of VLCFA precursors of C20–C24 and total cuticular wax load on wsl1 leaf blades and sheaths [9].

From a series of rice onion mutants showing small cabbage-like shoots or bladeless leaves, three lipid-related genes involved in the shoot apical meristem were fine-mapped [43]. As a fatty acid elongase (β-ketoacyl CoA synthase), ONI1 biochemically synthesizes VLCFAs in the outermost epidermal cell layer [10]. Similarly, ONI2 catalyzes the first step of elongation reactions of a carbon chain of VLCFAs. Disruption of ONI2 led to tiny shoots in which leaves were fused and ceased growing after germination [11]. ONI3 was shown to be the ortholog of *Arabidopsis* HOTHEAD, containing glucose-methanol-choline (GMC) oxidoreductase and NAD(P)-binding Rossmann-like domains. It is functionally involved in the biosynthesis of long-chain fatty acids. Compared to *oni1* and *oni2*, *oni3/mini1* showed more severe growth defects with fused neighboring organs and finally became lethal after germination [12,13].

Fibrillins (FBNs) are a conserved plastid-lipid-associated protein (PAP) family modulating lipid metabolism. In rice, Fibrillin 1 (OsFBN1), a chloroplast-localized protein, was found to specifically bind C18 and C20 fatty acids in vitro. Over-expression lines of OsFBN1 exhibited more tillers, short panicles, poor grain-filling percentage, and JA levels than wild-type and RNAi- silencing lines under heat stress. More interestingly, more plastoglobules, which are defined as a hub of lipid metabolism in the chloroplast [14], were observed in the over-expression lines, suggesting the essential role of rice *OsFBN1* in plastoglobule formation and plant growth [44].

Zhang et al. (2022) identified a rice semi-dwarf mutant, *semi-dwarf 38* (*sd38*), with significantly reduced cell length. They revealed that *SD38* encodes a fatty acid elongase responsible for the synthesis of C24:0 VLCFAs (VLCFAs). Exogenous application of VLCFA (C24:0) or ethephon could partially reverse the dwarf phenotype of *sd38* [15].

## 3. Lipids Control Rice Pollen Fertility

The outer layer of the anther, called the anther cuticle, and the pollen wall’s outer layer, known as pollen exine, are essential lipid layers for the development of the male reproductive organ [45]. The anther cuticle comprises cutin and cuticle wax, while the exine is composed of sporopollenin, a highly resistant biopolymer derived from fatty acids [46]. It is well established that defective lipid synthesis in the cuticle or exine leads to nonviable pollen and sterile plants. To date, numerous genes related to lipid metabolism that cause male sterility have been identified, and their functions have been elucidated in rice and other plant species [45].

*Wax-deficient anther1* (*Wda1*), designated as *OsGL1-5*, was cloned from a T-DNA insertional mutant showing pollen sterility and significant defects in the biosynthesis of VLCFAs in both layers. It was observed that epicuticular wax crystals were absent in the outer layer of the anther, and pollen exine formation was compromised in the mutant anthers [16]. Besides *Wda1*, rice acyl-CoA synthetase5 (OsACOS5) is also implicated in pollen exine development and fertility [17,18].

In plant cuticles, fatty alcohols synthesized by fatty acyl-CoA reductase are important fatty constituents [46]. DPW (Defective Pollen Wall) has been identified as a novel fatty acid reductase mediating the production of 1-hexadecanol. DPW and its *Arabidopsis* ortholog MS2 are functionally conserved, as their corresponding mutants display defected anther and degenerated pollen grains with an irregular exine [19,47]. In *dpw* anthers, cutin monomers were dramatically reduced with an altered composition of cuticular wax, soluble fatty acids, and alcohols. The work showcased the link between primary fatty alcohol synthesis, anther cuticle, and pollen sporopollenin biosynthesis in monocots and dicots [19].

GDSL esterases and lipases, featured by the conserved motif Gly-Asp-Ser-Leu, are a subfamily of hydrolytic/lipolytic enzymes [48]. Zhao et al. (2020) map-based cloned an endoplasmic reticulum-localized GDSL lipase gene *RMS2* from an irradiation-induced mutant population. *rms2* exhibited complete male sterility, while the vegetative growth was normal. Biochemically, *RMS2* possesses lipid hydrolase activity, while the dysfunction of *RMS2* led to significant changes in the content of 16 lipid components and numerous other metabolites. Interestingly, *RMS2* might be a key node in the rice male fertility regulatory network, since master male fertility regulators Undeveloped Tapetum1 (UDT1) and Persistent Tapetal Cell1 (PTC1) could activate its transcription [49].

A recent study revealed that lipid metabolism might be subject to epigenetic regulation [20]. *PEM1* (*pollen expressed MBD-like 1*) encoding a methyl-CpG-binding domain protein is involved in the pollen exine development. pem1 anthers became small and shrunken with 30% lower viable pollen grains when compared to the WT. Unlike many of the other male sterile mutants showing retarded tapetum degradation, *pem1* mutants had regular PCD progress of the tapetum. However, they exhibited abnormal Ubisch body formation, delayed exine occurrence, defective exines, and increased anther cuticles [20]. Additionally, a novel alpha integrin-like protein DPW3 (Defective Pollen Wall3) has been associated with pollen wall formation and fertility, although the detailed molecular mechanism requires further exploration [50].

Humidity-sensitive genic male sterility (HGMS) lines have great potential in hybrid rice breeding. However, HGMS-related genes have yet to be identified. Chen et al. (2020) isolated a rice HGMS mutant that exhibited male sterility at low humidity but was fully fertile at high humidity [21]. The causal gene *HUMIDITY-SENSITIVE GENIC MALE STERILITY 1* (*HMS1*) encodes a putative β-ketoacyl-CoA synthase. Through interacting with its co-factor HMS1-INTERACTING PROTEIN (HMS1I), HMS1 promotes the conversion of C24:0 and C26:0 fatty acids into C26:0 and C28:0 fatty acids on the pollen wall, which helps to protect pollen from dehydration under low-humidity conditions. In a yeast system, HMS1 and HMS1I were found to catalyze the biosynthesis of VLCFAs longer than C24. Notably, the authors developed an HGMS line using the *HMS1* gene with convertible pollen fertility under different humidity conditions, suggesting that *HMS1* can be potentially used for hybrid breeding in indica and japonica rice.

Unlike other reported *OsGLs* involved in drought resistance, *OsGL1-4/CER1* was identified as a critical regulator in male reproductive development. Ni et al. (2018) revealed that *OsGL1-4/CER1* functions in VLC alkanes biosynthesis, underpinning anther development and plastids differentiation. *OsGL1-4/CER1* transcription was robustly detected in the developed tapetum and bicellular pollen cells. Meanwhile, disruption of *OsGL1-4/CER1* significantly reduced the content of VLC alkanes (C25 and C27), finally leading to sterile pollen grains containing fewer amyloplasts [22]. A few years later, two independent groups both found that the pollen sterility of cer1 is attributed to the excessively fast dehydration at anthesis and defective adhesion and hydration under normal conditions, which could be recovered by artificial high humidity [51,52]. Ni et al. (2021) further specified that the lipid composition of tryphine is the primary reason for such humidity-sensitive genic male sterility, since the fertility could be recovered by co-pollination with mixed *OsCER1*Cas mutant and maize pollens [52].

## 4. Lipids Involves in the Regulation of Grain Yield

Enoyl-CoA hydratase (ECH) catalyzes the second step in fatty acid metabolism’s physiologically important beta-oxidation pathway [53]. A Chinese research group found that *NUMBER OF GRAINS 1* (*NOG1*) encoding an enoyl-CoA hydratase/isomerase regulates grain number per panicle without penalty on the panicle number and grain weight. A 12-bp indel in the promoter affected the transcription of *NOG1*, which finally contributed to the variations in total fatty acids and linolenic acid (LA, C18:3) among different varieties. In addition to the fatty acids, *NOG1* is involved in the biosynthesis of JA, since C18:3 is the synthetic precursor of JA. Introgression of *NOG1* in modern cultivars achieved up to a 25.8% increase in grain yield, showing the massive potential of lipid-related genes in crop genetic improvement [23].

Zhao et al. (2019) identified a seed-predominantly expressed microRNA, miR1432. The suppression of miR1432 facilitated grain filling rate and could increase grain yield up to 17.14% in field trials. In their search for the downstream targets of miR1432, the authors discovered that rice Acyl-CoA thioesterase (*OsACOT*) is a genuine target gene negatively regulated by miR1432-directed cleavage. *OsACOT* serves as a crucial enzyme in the fatty acid desaturation and elongation pathway, particularly in the conversion of the fatty acid 16:0 into 18:2. Intriguingly, over-expression of miR1432-resistant form of *OsACOT* mimicked miR1432 suppression lines and resulted in an increased yield of approximately 50% [24].

## 5. Grain Quality Is Affected by Lipid Contents and Components

Although rice grains possess a relatively low lipid content, the lipids in rice bran and grains exhibit a balanced composition of saturated (e.g., C16:0) and unsaturated (e.g., C18:1 and C18:2) fatty acids, which presents rice as a valuable and healthful oil source for human consumption [54]. Over the past few decades, substantial efforts have been dedicated to investigating the genetic underpinnings of rice lipid metabolism, with a particular emphasis on fatty acids. Employing a classic QTL mapping strategy, three QTLs, qRFC-1, qRFC-2, and qRFC-5, that accounted for 44% of the fat content variation were mapped in a doubled haploid population [55]. Later, two other groups did similar work, identifying nearly 20 QTLs contributing to grain lipid content. Nevertheless, no candidate genes underlying the QTLs were uncovered [56,57]. Ying et al. (2012) mapped 29 fatty acid content-associated QTLs, including eight QTLs that were repeatedly detected across multiple years and 11 QTLs harboring orthologs of key *Arabidopsis* lipid metabolism enzymes. Interestingly, a strong QTL for oleic (18:1) and linoleic (18:2) acids was associated with a rice ortholog of a gene encoding acyl-CoA: diacylglycerol acyltransferase (DGAT) and another for palmitic acid (16:0) mapped similarly to the acyl-ACP thioesterase (*FatB*) gene ortholog [58]. In 2021, a genome-wide association study was conducted in a diverse panel of 533 cultivated rice accessions and identified 46 loci controlling oil composition and concentration. Genes involved in lipid metabolism, namely *PAL6, LIN6, MYR2*, and *ARA6*, are significant contributors to the natural variance in oil composition within rice subpopulations. Notably, *qPAL6* exhibits a substantial effect, explaining 49.86% and 57.44% of the phenotypic variance in the C16:0 composition. Similarly, qLIN6 accounts for 28.56% and 55.31% of the phenotypic variance in C18:2 conversion from C18:1. Furthermore, *qMYR2* encodes a myristoyl-ACP thioesterase responsible for the hydrolysis of specific acyl-ACPs, leading to the production of saturated fatty acids. Disruption of *qMYR2* results in lowered C14:0 and C16:0 but increased C18:2. Additionally, *qARA6*, annotated as a 3-ketoacyl-CoA synthase, modulates the biosynthesis of very-long-chain fatty acids (C20–C22). Functional impairment of *qARA6* significantly reduces the C20:0, C20:1, C20:2, and C22:0 components [59].

The economic significance of rice oil has led to the implementation of various strategies aimed at enhancing the lipid content and healthy fatty acid components through genetic means. One key strategy involves boosting lipid synthesis efficiency by utilizing lipid rate-limiting enzymes or regulator genes from rice or other plant species. WRINKLED1 (WRI1) is a master regulator of seed oil biosynthesis in *Arabidopsis*. Over-expression of *AtWRI1* in rice increased fatty acid content by 30–40% in vegetative organs but decreased fatty acid content in the endosperm, whereas the endosperm-specific expression of *CoWRI1* from coconut showed minor effects on rice oil content [25,60]. Over-expression of *RAG2*, a member of 14-to-16-kDa α-amylase/trypsin inhibitors in rice, gave rise to a 10–30% increase in lipid content and higher storage protein levels [26].

However, given that the overall carbon flux to starch is dominant over lipid biosynthesis, it is not surprising that minor effects could be observed when only lipid biosynthesis is enhanced [31]. As suggested by a report on *Chlamydomonas reinhardtii*, in which lipid content increased by 3.5-fold when an essential starch synthesis gene was knocked out [61], blocking the starch biosynthesis might be an efficient strategy to divert the carbon flux to lipid biosynthesis. Indeed, Wei et al. (2017) knocked out an ADP-glucose pyrophosphorylase (*AGP*) large subunit gene to have the oil content in rice grains over 10% [27].

The protein oleosin is notably abundant in the oil bodies of plant seeds, and it serves a crucial function in the accumulation of lipids. By introducing two soybean oleosin genes that encode 24 kDa proteins in rice, under the control of an embryo-specific rice promoter *REG-2*, the transgenic rice showed more oil bodies in smaller size and a significant increase in seed lipid content. Specifically, the lipid content increased by 36.93% and 46.06% compared to the control, while the fatty acid profiles of triacylglycerols remained unchanged [28]. In addition, modifying *OsROS1* and *OsMitssb1* significantly increased the thickness of rice aleurone layer cells, which could finally lead to an over-48% increase in grain oil content, representing an alternative strategy in rice lipid metabolic engineering [29,62].

In pursuit of achieving a balanced carbon flux between carbohydrates and lipids in rice, Izadi-Darbandi et al. (2020) undertook the metabolic engineering of rice grains by integrating *Arabidopsis* genes such as *AtWRI1*, *AtOle*, diacylglycerol acyltransferase (*AtDGAT*), and phospholipid:diacylglycerol acyltransferase (*AtPDAT*) to facilitate the final step of TAG assembly. This intervention substantially increased TAG, oleic acid, palmitic acid, and total oil contents by 26%, 28%, 27%, and 70%, respectively [30]. In 2023, Liu et al. published a significant case of multigene engineering aimed at augmenting lipid content in rice grains. The goal was achieved by the concurrent knock-out of two rice genes, *AGPL2* and *Mtssb*1, and the endosperm-specific expression of *DGAT1* from *Arabidopsis*. These modifications effectively redirected carbon flux from starch to lipid, resulting in a substantial increase in oil biosynthesis, elevating the lipid content from approximately 2% to around 12%. Meanwhile, the fatty acid compositions were significantly altered with more beneficial components such as GLA. A lipid content of 12% is very close to oil crops like soybean, which represented the highest lipid level ever reported in rice or other starchy-type grains [31]. This work highlighted a practical approach for genetically improving oil contents in rice and other crops with starchy grains.

Rice oil comprises approximately 18% palmitic acid, 36% oleic acid, and 37% linoleic acid. Linoleic acid is particularly susceptible to non-enzymatic oxidation due to its oxidative instability [63]. The high oleic acid content in rice has consistently been a desirable characteristic due to its enhanced stability and beneficial nutritional profile. It is well known that the microsomal omega-6 fatty acid desaturase (FAD2) catalyzes the conversion of oleic acid to linoleic acid in seed oil. In *Arabidopsis* and other cops, knock-out or knock-down of *FAD2* successfully produced high oleic and low linoleic oil seeds [64,65]. In rice, the suppression of *FAD2-1*, one of the transcripts of *FAD2* predominantly expressed in seeds, through RNA interference (RNAi), has been observed to lead to an increase in oleic acid and a reduction in linoleic and palmitic acids in the grains. This finding suggests that *OsFAD2-1* represents a viable target for enhancing the genetic profile of fatty acids in rice [32,33]. In an attempt to enhance the α-linolenic acid (ALA) content and to reduce the ratio of linoleic acid (LA) to ALA, Liu et al. (2012) systematically introduced six ω-3 (Δ-15) fatty acid desaturase (FAD) genes from rice and soybean into the rice. The ALA contents in the seeds of *GmFAD3-1* and *OsFAD3* over-expression lines exhibited a consistent increase of 23.8- and 27.9-fold across multiple generations. Comparative analysis revealed that the endosperm-specific promoters GluC and REG outperformed the constitutive maize Ubi-1 promoter in enhancing ALA levels in rice embryos and bran [34,66].

## 6. Lipids Regulate Seed Longevity in Rice

The longevity and viability of seeds are closely associated with the presence of lipid peroxidation products, specifically malondialdehyde and acetaldehyde. These products can lead to cell damage and intoxication by reacting with macromolecules. Linolenic acid (LNA) and LA are the most significant polyunsaturated fatty acids. The peroxidation of LNA and LA is typically linked to the breakdown of cell structure, the formation of cytotoxic products, and the release of volatile decomposition products, all of which contribute to the deterioration of rice seed longevity. Lipoxygenases, a conserved family of fatty acid dioxygenases, facilitate the addition of oxygen to polyunsaturated fatty acids such as LNA and LA. Among the three embryo-derived LOX isoenzymes, LOX3 is at a dominant level and has been linked to fatty acid peroxidation during seed storage and stored grain quality [67]. Xu et al. (2015) found that LOX3 is capable of producing 9-hydroperoxyoctadecadienoic acid (9-HPOD) using LA as substrate. Meanwhile, suppressing *LOX3* expression in rice endosperm successfully extended rice seed longevity under either artificial aging or natural aging conditions, whereas the major agronomic traits remain unchanged [35]. In addition, the knock-down of *LOX3* could also effectively protect the functional component β-carotene from deterioration in the carotenoid-enriched golden rice [68].

## 7. Lipids Related to Abiotic Stress Response in Rice

Cuticular wax on the leaf aerial surfaces is the outermost diffusion barrier against the uncontrolled loss or uptake of water and gases [69]. To most plants, cuticular waxes are complex mixtures of primarily VLCFAs, hydrocarbons, alcohols, flavonoids, and so on [46], though the proportions of the major constituents may vary from each other. Because the defective accumulation or altered composition of wax could be visually detected, screening of cuticular wax mutants like eceriferum (cer) in *Arabidopsis* and Glossy1 in maize has been carried out for decades [46,70]. In rice, Islam et al. (2009) systematically identified 11 homologs of maize GL1, designated as *OsGL1-1* to *OsGL1-11*. According to phylogenetic analysis, most of the OsGL1s could be induced by abiotic stresses and might be involved in cuticular or epicuticular wax biosynthesis [71]. At least, *osgl1-1*, *osgl1-2*, *osgl1-3*, and *osgl1-6* have been found to participate in drought stress response in rice. Compared to the WT, the *osgl1-2* exhibited less wax crystallization, a thinner cuticular layer, and reduced total cuticular wax amount, leading to higher sensitivity to drought stress at the reproductive stage. In contrast, the over-expression lines showed the opposite phenotypes [71]. *OsGL1-1* was annotated as a sterol desaturase or short-chain dehydrogenase/reductase. *OsGL1-3* protein has a conserved fatty acid hydroxylase domain (FAH domain) and a WAX2 C-terminal domain, while *OsGL1-6* gene is a member of the fatty aldehyde decarbonylase gene family. The *osgl1-1*, *osgl1-3* and *osgl1-6* mutants showed very similar phenotypes to *osgl1-2,* with decreased cuticular wax deposition, thinner cuticular membranes, and hypersensitivity to drought stress [72,73,74].

It was found that *osa-miR1848* regulates *OsWS1*, a membrane-bound O-acyl transferase gene member, to determine wax biosynthesis [36]. *OsWS1* and *osa-miR184*8 exhibited a negative time-and-spatial co-expression pattern, especially under water-deficit treatment, suggesting antagonistic roles of the two factors. Compared to the control, *OsWS1* over-expression lines displayed 35% higher VLCFA contents, denser wax papillae around the stoma, and a higher survival rate upon water-deficit treatment. Conversely, *OsWS1*-RNAi and *osa-miR1848* over-expression plants exhibited opposing changes.

*OsWIH2* is another reported fatty acid synthesis enzyme that positively regulates rice drought resistance by alleviating water loss and reactive oxygen species (ROS) accumulation and altering wax content [37]. *OsWIH2* works as a complex with HOTHEAD (HTH), which is an α-alcohol dehydrogenase catalyzing the biosynthesis of long-chain α-,ω-dicarboxylic fatty acids (LCFAs) in the cutin and wax biosynthesis pathway [75]. Meanwhile, drought-inducible transcription factor OsbHLH130 could activate the transcription of *OsWIH2*, thus forming a bHLH130-OsWIH2-HTH regulatory module in cuticular wax biosynthesis and drought response.

Besides the wax or cuticle synthesis enzymes, several genes involved in the regulation of wax biosynthesis and drought tolerance have been identified. These include the transcription factor *CFL1*, *OsWR2*, and *DWA1* [39,76,77,78]. Wang et al. (2018) reported that DROUGHT HYPERSENSITIVE (DHS) negatively regulates wax biosynthesis by promoting the turnover of an HD-ZIP IV family member ROC4, thereby influencing drought tolerance [38]. Over-expression of DHS significantly reduced the cuticular wax contents and conferred drought hypersensitivity. As a RING-type E3 ligase, DHS functions in mediating the ubiquitination of ROC4 in vivo, subsequently leading to its degradation via the ubiquitin/26S proteasome-mediated pathway. Notably, the DHS-ROC4 module has been identified as directly targeting Os-BDG, a putative regulator of wax biosynthesis. Thus, this study elucidates a precisely regulated molecular mechanism that links wax biosynthesis with drought stress response in rice, offering valuable insights for developing drought-tolerant rice cultivars.

Xiang et al. (2022) identified *Salt Tolerance and Heading Date 1* (*STH1*) from the African rice species Oryza glaberrima. *STH1* encodes an α/β hydrolase family member primarily localized in the major sites of peroxisomal β-oxidation, including the cytoplasm and mitochondria. It catalyzes the hydrolytic degradation of unsaturated fatty acids to maintain fatty acid metabolic homeostasis and membrane fluidity, thus controlling rice salt, heat, and osmotic tolerance. Correspondingly, rice seedlings showed a more significant growth acceleration in terms of shoot length when grown on a mannitol-containing medium supplemented with the fatty acid 11-eicosenoic acid, implying that metabolic flux redirection by blocking fatty acids from degradation can enhance plant tolerance to various abiotic stresses. *STH1HP46* is found to be a rare allele in modern Asian cultivars, which shows great promise in boosting grain yield under salt stress [79].

Very few favorable thermo-tolerance genes have been identified, mostly working through reactive oxygen species scavenging, cytotoxic protein elimination, and unfolded protein renaturation [80,81]. Kan et al. (2022) revealed a novel “TT2-SCT1-OsWR2” thermal tolerance mechanism linking G protein, Ca2+ sensing, and wax metabolism [82]. Known as a Gγ subunit protein, *TT2* was cloned as a major QTL contributing to heat tolerance. Heat-treated plants carrying a weak allele of *TT2* had enhanced thermo-tolerance, exhibiting less-severe reductions in wax content compared with plants with the normal *TT2* allele. The authors have validated that the heat signal increases cytosolic Ca2+, which subsequently undergoes conversion into an active form facilitated by CaM. Following this, CaM binds to SCT1 to repress its activity, thereby decreasing the transcription of *OsWR2*, the pivotal regulator of wax biosynthesis [39]. Consequently, this perturbation results in reduced wax content and induces thermo-sensitivity. Field trial experiment results showed that *TT2* governs rice thermo-tolerance in both vegetative and reproductive stages. Under heat stress, NIL-TT2HPS32 plants significantly outperformed NIL-TT2HJX plants with 54.7% higher grain yield, showing promising potential in breeding thermo-tolerant rice varieties.

## 8. Rice Biotic Stress Response Is Mediated by Lipids

Microdomains, also called membrane rafts, are small, heterogeneous, liquid-ordered domains composed mainly of sphingolipids and sterols [83]. Since plant defense-related proteins primarily congregate in plasma membrane microdomains, innate immunity is associated with microdomains’ functions. By knocking down two fatty acid 2-hydroxylases catalyzing the 2-hydroxylation of sphingolipid fatty acids, Nagano et al. (2016) obtained *fah1 fah2* genetic lines harboring fewer microdomains and higher susceptibility to rice blast fungus infection. *Os-RbohB* and *Os-RbohH* are microdomains-located NADPH oxidases involved in the pathogen defense process. They revealed that microdomains are required for the dynamics of the GTPase Rac1 and respiratory burst oxidase homologs (Rbohs) in response to chitin elicitors, regulating chitin-induced immunity through ROS signaling mediated by the Rac1-RbohB pathway [40].

Many plant phosphatidylinositol phosphates, particularly PtdIns(4,5)P2, have been demonstrated to be disease-susceptibility factors [84]. From a rice lesion mimic mutant with broad-spectrum disease resistance, Sha et al. (2023) cloned a cytidine diphosphate diacylglycerol synthase gene *RBL1* involved in phospholipid biosynthesis. *rbl1* had drastically reduced the amount of phosphatidylinositol and its derivative PtdIns(4,5)P2, which is enriched in the biotrophic interfacial complex and extra-invasive hyphal membrane structure associate with effector secretion and fungal infection. Intriguingly, the authors generated a valuable allele named *RBL1Δ12*, which confers broad-spectrum disease resistance to 10 *M. oryzae* field strains, five *Xoo* strains, and two rice false smut *Ustilaginoidea virens* strains in model rice variety backgrounds. At the same time, no yield penalty was observed in multiple field trials [85]. The work demonstrated the feasibility of balancing rice growth and immunity via genetically editing lipid-related genes.

In addition to its functions in seed longevity, Lipoxygenase 3 (LOX3) may negatively influence rice blast disease resistance. *lox3* mutants grow normally with high levels of methyl-linolenate and reactive oxygen species (ROS). After *M. oryzae* infection, *lox3* plants exhibited serious blast symptoms and reduced defense response, accompanied by increased ROS-mediated cell death. Moreover, exogenous JA repressed hyphal expansion and ROS-mediated cell death, finally alleviating blast symptoms in the mutant [86].

A recent publication described the functional characterization of *OsGELP77*, a major QTL contributing to broad-spectrum resistance in rice. *OsGELP77* is annotated as an ER-localized, GDSL-type lipase with typical lipase activity. Based on its pathogen- and JA-inducible expression pattern, *OsGELP77* was found to positively regulate resistance to various pathogens, possibly through modulating glycerolipid metabolism, glycerophospholipid metabolism, and sphingolipid metabolism as well as JA homeostasis. In addition, *OsWRKY45* may work upstream of *OsGELP77* to trigger rice immunity. Elevated expression of *OsGELP77* or pyramiding of a natural elite haplotype *OsGELP77*Hap3 significantly increased resistance to various pathogens to achieve higher yield [41].

Oxylipins are a big family of lipid compounds well documented for their function in insect resistance in rice. *LOX11* (Lipoxygenase-11)/*OsRCI-1* is a chloroplast-localized protein gene in rice showing induced expression by brown planthopper (BPH) infestation. Over-expression of *OsRCI-1* elevated the levels of JA, jasmonate-isoleucine, and trypsin protease inhibitors, which imposed decreased colonization, fecundity, and mass of the BPH insects when compared with the WT. Moreover, the decreased attractiveness to BPH and enhanced attractiveness to the parasitoid of *oeRCI* plants correlated with higher levels of BPH-induced 2-heptanone, 2-heptanol, and 2-heptanone. These results indicate that *OsRCI-1* is involved in herbivore-induced JA bursts and plays a role in plant defense [42].

## 9. Perspectives

This comprehensive review delineates recent progress in elucidating the genetic underpinnings and physiological functions of lipids in rice, a model species for monocots and cereals. Despite the aforementioned advancements, our knowledge of the biological roles of lipids in rice remains relatively rudimentary and fragmented compared to that in *Arabidopsis* and other oil crops. The bulk of published rice cases hinge on robust correlations between phenotype and variations in lipid content or components. However, the molecular mechanisms underlying the relationship between lipid variation and phenotypic diversity remains largely unexplored. One example that has been thoroughly documented is the role of *STH1* in catalyzing the hydrolytic degradation of unsaturated fatty acids, which is essential for maintaining membrane fluidity [79]. Meanwhile, a large number of rice lipid-related genes have been characterized via a forward genetics approach utilizing randomly collected mutants and map-based cloning, which resulted in a fragmented comprehension of the rice lipid regulation network. Thus, there is a need to allocate additional resources to systematically explore the gene families associated with lipid biosynthesis, metabolism, and signaling. Finally, as has been shown in a few cases in rice [87], the advancement of new mass spectrometric techniques in lipidomic research has the potential to significantly enhance the identification of novel lipid components within the pathways and networks of cellular lipids in rice biological systems.

## Figures and Tables

**Figure 1 ijms-25-09046-f001:**
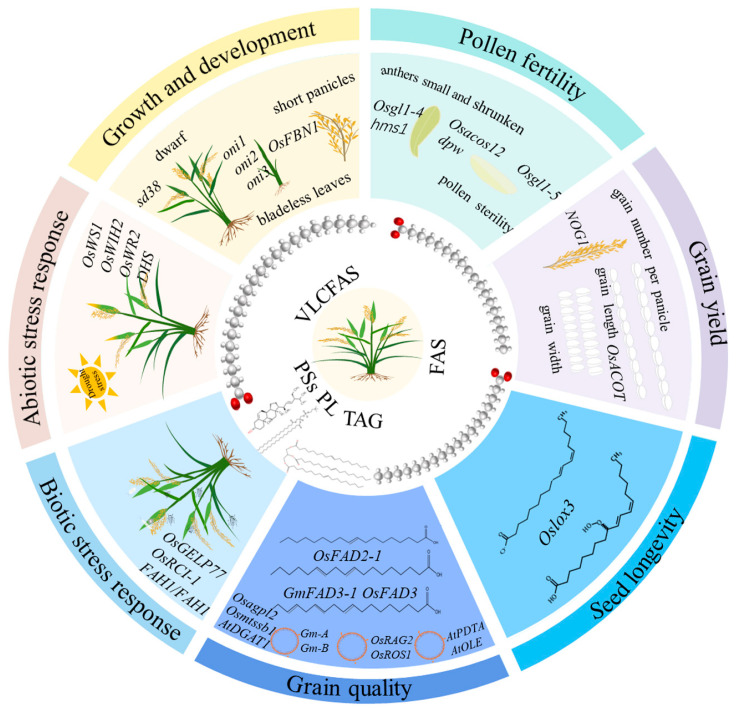
Significant roles of lipids in rice. FAS: fatty acids; PSs: phytosterols; PL: Phospholipid; TAG: triacylglycerol; VLCFAS: very-long-chain fatty acids.

**Table 1 ijms-25-09046-t001:** A summary of the lipid-related genes and their functions in rice.

Gene	LOC ID	Annotation	Phenotype Description	References
** *WSL1* **	*LOC_Os06g39750*	β–ketoacyl CoA synthase	The mutants had reduced growth, leaf fusion, sparse wax crystals, and low fertility.	[9]
** *ONI1* **	*LOC_Os03g08360*	β–ketoacyl CoA synthase	The mutants produced small shoots and aberrant outer epidermal cells and ceased to grow after germination.	[10]
** *ONI2* **	*LOC_Os10g28060*	β–ketoacyl CoA synthase	The mutants had tiny shoots, fused leaves and ceased growing after germination.	[11]
** *ONI3* **	*LOC_Os09g19930*	Long-chain fatty acid ω-alcohol dehydrogenase	The mutants had severe growth defects with fused neighboring organs and finally became lethal.	[12,13]
** *OsFBN1* **	*LOC_Os09g04790*	Probable plastid-lipid-associated protein	Over-expression of *OsFBN1* exhibited more tillers, short panicles, poor grain-filling percentage, and JA levels under heat stress.	[14]
** *SD38* **	*LOC_Os10g33370*	β–ketoacyl CoA synthase	The mutants’ cell length, 1000-grain weight, and seed setting rate were significantly reduced.	[15]
** *OsGL1-5* **	*LOC_Os10g33250*	Glossy1-homologous gene	The mutants showed pollen sterility and significant defects in the biosynthesis of VLCFAs.	[16]
** *OsACOS12* **	*LOC_Os04g24530*	Acyl-CoA synthetase	The mutants had normal vegetative growth but white anthers that were shorter and smaller and produced no mature pollen grains	[17,18]
** *DPW* **	*LOC_Os03g07140*	Fatty Acyl Carrier Protein Reductase	The mutants display defected anther and degenerated pollen grains with an irregular exine	[19]
** *PEM1* **	*LOC_Os02g09920*	Methyl-CpG-binding domain family member	*pem1* anthers became small and shrunken, with 30% lower viable pollen grains	[20]
** *HMS1* **	*LOC_Os03g12030*	β-ketoacyl-CoA synthase	The mutants that exhibited male sterility at low humidity but were fully fertile at high humidity	[21]
** *OsGL1-4* **	*LOC_Os02g40784*	Glossy1-homologous gene	The mutants significantly reduced the content of VLC alkanes (C25 and C27), finally leading to sterile pollen grains	[22]
** *NOG1* **	*LOC_Os01g54860*	Enoyl-CoA hydratase/isomerase	*NOG1* can significantly increase the grain yield of commercial high-yield varieties.	[23]
** *OsACOT* **	*LOC_Os04g35590*	Acyl-CoA thioesterase	Over-expression of the miR1432-resistant form of *OsACOT* resulted in an increased yield of approximately 50%.	[24]
** *AtWRI1* **	*AT3G54320*	AP2/EREBP transcription factor	Over-expression of *AtWRI1* in rice increased fatty acid content by 30–40% in vegetative organs but decreased fatty acid content in the endosperm.	[25]
** *RAG2* **	*LOC_Os07g11380*	α-amylase/trypsin inhibitor	Over-expression of *RAG2* gave rise to a 10–30% increase in lipid content and higher storage protein levels.	[26]
** *OsAGPL2* **	*LOC_Os01g44220*	ADP-glucose pyrophosphorylase large subunit	The oil content of mutant grains reached over 10%.	[27]
** *Gm-A* ** ** *Gm-B* **	*U09118* *U09119*	24 kDa oleosin	Transgenic rice showed greater numbers of smaller oil bodies, and lipid content increased by 36.93% and 46.06%, respectively	[28]
** *OsROS1* **	*LOC_Os01g11900*	DNA demethylase	The number of cell layers in the anthers increased significantly, while the oil content of the rice increased by 48%, and the vitamin E content doubled	[29]
** *AtPDAT* ** ** *AtDGAT1* ** ** *AtWRI1* ** ** *AtOLE* **	*AT5G13640* *AT2G19450* *AT3G54320* *AT4G25140*	Phospholipid:diacylglycerol acyltransferase Diacylglycerol acyltransferase AP2/EREBP transcription factor Oleosin1	Increased TAG, oleic acid, palmitic acid, and total oil contents by 26%, 28%, 27%, and 70%	[30]
** *OsAGPL2* ** ** *OsMTSSB1* ** ** *AtDGAT1* **	** *LOC_Os01g44220* ** ** *LOC_Os05g43440* ** ** *AT2G19450* **	ADP-glucose pyrophosphorylase large subunit Mitochondrion-targeted single-stranded DNA-binding protein Diacylglycerol acyltransferase	The lipid content increased from approximately 2% to around 12%,	[31]
** *OsFAD2-1* **	*LOC_Os02g48560*	ω-6 fatty acid desaturase	Mutant showed an increase in oleic acid and a reduction in linoleic and palmitic acids in the grains	[32,33]
** *GmFAD3-1* ** ** *OsFAD3* **	*GLYMA_14G194300v4* *LOC_Os12g01370*	microsomal omega-3-fatty acid desaturase ω-3 fatty acid desaturase gene	GmFAD3-1 and OsFAD3 over-expression lines consistently increased 23.8- and 27.9-fold across multiple generations.	[34]
** *OsLOX3* **	*LOC_Os03g49260*	lipoxygenase gene	Suppressing *LOX3* expression in rice endosperm successfully extended rice seed longevity	[35]
** *OsWS1* **	*LOC_Os04g40590*	Membrane-bound O-acyl transferase gene	*OsWS1* over-expression lines displayed 35% higher VLCFA contents, denser wax papillae around the stoma, and a higher survival rate upon water-deficit treatment.	[36]
** *OsWIH2* **	*LOC_Os03g31679*	cysteine-rich and transmembrane domain-containing protein WIH2	Over-expression of *OsWIH2* positively regulate rice drought resistance by alleviating water loss, reducing reactive oxygen species (ROS) accumulation, and altering wax content	[37]
** *DHS* **	*LOC_Os02g45780*	RING-type E3 ubiquitin ligase	Over-expression of DHS significantly reduced the cuticular wax contents and conferred drought-hypersensitive.	[38]
** *OsWR2* **	*LOC_Os06g40150*	Ethylene response factor	Over-expression of *OsWR2* increased the total cuticular wax in leaves and panicles, decreased water loss, and enhanced drought resistance.	[39]
** *OsFAH1* ** ** *OsFAH2* **	*LOC_Os12g43363* *LOC_Os03g56820*	fatty acid 2-hydroxylase	*fah1fah2* harbors fewer microdomains and higher susceptibility to rice blast fungus infection	[40]
** *OsGELP77* **	*LOC4340180*	GDSL esterase/lipase gene	Elevated expression of *OsGELP77* or pyramiding of a natural elite haplotype significantly increased resistance to various pathogens	[41]
** *OsLOX11/OsRCI-1* **	*LOC_Os12g37260*	lipoxygenase gene	Over-expression of *OsRCI-1* elevated the levels of JA, jasmonate-isoleucine, and trypsin protease inhibitors, which decreased colonization, fecundity, and mass of the BHP insects	[42]

## Data Availability

Not applicable.

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
