# Peer review of "Biological Roles of Lipids in Rice"

_ijms, 2024, doi:10.3390/ijms25169046_

Round 1

Reviewer 1 Report

Comments and Suggestions for Authors

The topic of the review is novel, and this review is very timely. However, there are areas where the review can be further improved. I have several comments for the authors that should be addressed to make the review more informative and engaging for readers. My comments are as follows:

Lines 44-45 and Lines 54-55: Please support the statements with references. There are several areas in the manuscript that need to be supported by appropriate references. The authors are requested to review the entire manuscript and provide appropriate citations wherever needed.

The authors could provide a better justification for why this review is needed and how it differs from existing reviews on the topic.

The titles for the different sections are not appropriately assigned. After the introduction, the authors directly move to "growth and development," among other titles, which do not clearly indicate what is discussed in each section. The authors should provide clearer and more descriptive titles for all sections.

Line 71-81: In this paragraph, the authors discuss onion mutants but cite references related to rice. The authors should carefully revise this paragraph to ensure that the references are relevant and correctly support the content.

One major issue with this review is that the authors have compiled previous information without highlighting what is lacking in each research area and what potential future research areas could be explored. I highly recommend that the authors better synthesize the available information and clearly identify research gaps in each section. Alternatively, the authors could include a separate section specifically dedicated to highlighting research gaps and potential future research areas.

Comments on the Quality of English Language

The manuscript can be benefitted from a thorough review for English language improvement. There are several instances where sentence structure, word choice, and grammar could be refined for better clarity and flow.

Author Response

Comment: Lines 44-45 and Lines 54-55: Please support the statements with references. There are several areas in the manuscript that need to be supported by appropriate references. The authors are requested to review the entire manuscript and provide appropriate citations wherever needed.

Response: Thank you very much for the suggestive comments. We thoroughly investigated the reference issue and included more citations to support our statement.

Comment: The authors could provide a better justification for why this review is needed and how it differs from existing reviews on the topic.

Response: In short, compared to the existing ones, we focused on rice, other than oil crops, to summarize the scientific advances in the biological roles of lipids. As we mentioned in L42-55, lipids are essential to plants. Previous efforts on lipids' functions were mainly put on oil crops, while those in rice have been largely ignored due to the relatively low lipid contents in rice. To the best of our knowledge, no comprehensive review is currently available on the biological roles of lipids in rice, even though rice is one of the world's most important crops and a model species for plant molecular biology.

Comment: The titles for the different sections are not appropriately assigned. After the introduction, the authors directly move to "growth and development," among other titles, which do not clearly indicate what is discussed in each section. The authors should provide clearer and more descriptive titles for all sections.

Response: We made the subtitle a complete sentence.

Comment: Line 71-81: In this paragraph, the authors discuss onion mutants but cite references related to rice. The authors should carefully revise this paragraph to ensure that the references are relevant and correctly support the content.

Response: We are actually talking about rice onion mutants here. To avoid confusion, we specified that they are in rice in the text.

Comment: One major issue with this review is that the authors have compiled previous information without highlighting what is lacking in each research area and what potential future research areas could be explored. I highly recommend that the authors better synthesize the available information and clearly identify research gaps in each section. Alternatively, the authors could include a separate section specifically dedicated to highlighting research gaps and potential future research areas.

Response: We included a perspective section to highlight the research gap and critical issues in the future. Once again, the authors sincerely appreciate your efforts to improve our work.

Reviewer 2 Report

Comments and Suggestions for Authors

In the manuscript entitled “Biological roles of lipids in rice”, authors reviewed the biological roles of lipids in rice. They give a wide overview of the advancements in understanding lipids' genetic foundation and the physiological function of different lipids components. The manuscript is in the aims of the International Journal of Molecular Sciences however it needs to be polished before publication. I recommend a minor revision.

General considerations

I would like to stress the importance of references in the manuscript, especially for a Review. Please, add references in L24-L46. Only one reference in a lot of rows of lines is not acceptable for a review or for a scientific manuscript in general. 

Furthermore, please use the italic when needed. For example, in L223 “Chlamydomonas reinhardtii” is not in italics. Check all of this in all the manuscripts.

Check for typos, for example in L49 “Lipids”.

L51-L55: “Despite…processes.” Can you add some references to that? 

L95-L96: “The outer…organ”. Add references.

L127-L128: “A recent study…” Which one in the same row? Add reference.

L183-L186: There is a correlation or a study between the consumption of rice and a “healthful and nutrient source” for human consumption? There are only 2 types of fatty acids in rice? Remove “(C16:0) and unsaturated (C18:1 and C18:2) fatty acids” it is an incomplete statement. 

Author Response

Comment: I would like to stress the importance of references in the manuscript, especially for a Review. Please, add references in L24-L46. Only one reference in a lot of rows of lines is not acceptable for a review or for a scientific manuscript in general. 

Response: Thank you very much for the suggestive comments. Per your suggestions, we included 5 more citations to support the statement of lipid biological functions, industrial usage, and reviews in crops.

Comment: Furthermore, please use the italic when needed. For example, in L223 “Chlamydomonas reinhardtii” is not in italics. Check all of this in all the manuscripts.

Response: Corrected as suggested.

Comment: Check for typos, for example in L49 “Lipids”.

Response: We carefully checked all the typos and made corrections.

Comment: L51-L55: “Despite…processes.” Can you add some references to that? 

Response: The reference of Li et al, 2022 was included.

Comment: L95-L96: “The outer…organ”. Add references.

Response: The reference of Wan et al., 2020 was included.

Comment: L127-L128: “A recent study…” Which one in the same row? Add reference.

Response: The reference of Song et al, 2022 was included.

Comment: L183-L186: There is a correlation or a study between the consumption of rice and a “healthful and nutrient source” for human consumption? There are only 2 types of fatty acids in rice? Remove “(C16:0) and unsaturated (C18:1 and C18:2) fatty acids” it is an incomplete statement. 

Response: We specified that rice is a valuable and healthful oil source for human consumption, and the C16:0, C18:1, and C18:2 are just a few examples of the SFAs and USFAs, respectively. We merged the two sentences as one to present a better logic flow. Again, the authors sincerely appreciate your efforts to improve our work.

Round 2

Reviewer 1 Report

Comments and Suggestions for Authors

The authors have addressed my comments. I believe the manuscript can now be accepted for publication.